# Managing the Microbiome: How the Gut Influences Development and Disease

**DOI:** 10.3390/nu13010074

**Published:** 2020-12-29

**Authors:** Noah Weinstein, Brandon Garten, Jonathan Vainer, Dulce Minaya, Krzysztof Czaja

**Affiliations:** Department of Veterinary Biosciences and Diagnostic Imaging, University of Georgia, Athens, GA 30602, USA; Noah.Weinstein@uga.edu (N.W.); Brandon.Garten@uga.edu (B.G.); Jonathan.Vainer@uga.edu (J.V.); Dulce.MinayaCaba@uga.edu (D.M.)

**Keywords:** microbiota, dysbiosis, gavage, fecal transplant, microbiome manipulation, germ free, animal models, cecum

## Abstract

The microbiome lies at the forefront of scientific research, as researchers work to uncover its mysterious influence on human development and disease. This paper reviews how the microbiome is studied, how researchers can improve its study, and what clinical applications microbiome research might yield. For this review, we analyzed studies concerning the role of the microbiome in disease and early development, the common methodologies by which the microbiome is researched in the lab, and modern clinical treatments for dysbiosis and their possible future applications. We found that the gut microbiome is essential for proper development of various physiological systems and that gut dysbiosis is a clear factor in the etiology of various diseases. Furthermore, we found that germ-free animal models and microbiome manipulation techniques are inadequate, reducing the efficacy of microbiome research. Nonetheless, research continues to show the significance of microbiome manipulation in the clinical treatment of disease, having shown great promise in the prevention and treatment of dysbiosis. Though the clinical applications of microbiome manipulation are currently limited, the significance of dysbiosis in the etiology of a wide array of diseases indicates the significance of this research and highlights the need for more effective research methods concerning the microbiome.

## 1. The Microbiome: Not New, Just Newly Defined

The human “microbiome”, as it is often referred to today, is the collection of 10–100 trillion symbiotic microbes that live on and within the human body [1]. This microbiome represents the direct link between all living things on the planet and presents a unique avenue for researchers to understand this living link. As early as 1683, Dutch scientist Antonie van Leeuwenhoek noted in his own oral and fecal samples, “little living animalcules, very prettily a-moving” [2,3,4]. The scientific community, building upon these centuries of research, has now come to understand that the human microbiome plays a profound role in human development and growth, and in adult health. Microbiota can be found on nearly every surface of the human body, and diversely influence critical events in early development, the digestion of food, and the onset of disease [5]. While the progress made since Leewenhoek’s original discovery has been exponential, researchers have only begun to understand the role the human microbiome plays in our daily function.

## 2. Animals Models

Rodent models have found popularity in gut microbiome research (Figure 1), having been subjected to a wide range of methods to change their microbiota. A popular field applied to research in rodents is gnotobiology, the study of animals that are completely devoid of microorganisms or those that contain a defined and known set of microbial species. In such experiments, the mice are either germ-free (GF) and thus contain no species at all, monoxenic with just one known colonized species, or have defined microbiota with all species known [6]. There is a clear advantage in using GF mice when compared with a mono-associated (monoxenic) or defined microbiota. However, the microbiome plays a crucial role in development and GF mice exhibit related physiological consequences, including underdeveloped immune systems. A study found that GF 129S6 and GF C57BL/6 mice contained nearly no Th17 cells, and GF Swiss–Webster mice completely lacked any presence of Th17 cells [7]. Transplanting fecal matter from normal Jackson B6 or Taconic B6 mice into C57BL/6 mice via oral gavage resulted in normal Th17 cell levels after 10–12 days. However, these abnormalities in the development of gnotobiotic models, as well as their high cost, have limited their use in research.

There are two major ways to deplete the gut microbiome in mice to study its relationship with disease and physiology. The first is the previously mentioned GF model. The other model is an antibiotic treatment that can destroy the entire microbiome, often referred to as antibiotic-induced microbiome depletion (AIMD). The antibiotic model helps to avoid several issues that arise in GF models, the most significant advantage being that mice can undergo a normal colonization and full postnatal development following birth [8]. Allowing the rodents to fully develop before depleting the gut microbiome ensures that many of the immune system complications that arise in GF models do not interfere with the study.

The most popular method to manipulate the gut microbiome that does not involve depletion is oral gavage-assisted fecal transplants (Figure 1), in which a large tube is placed through the mouth and into the digestive tract of the rodent to deliver the fecal matter. With the ability to accurately control the dosage and placement of fecal matter and other compounds, the stressful side effects of gavage are often overlooked. Researchers have studied the process of oral gavage on its own to precisely identify how it directly impacts the physiological state in a rodent. A study reported that daily manual restraint and oral gavage in mice did not have a negative impact in body weight change, nylon bone toy weight change, fecal corticosterone levels, and response to the anxiety drug chlordiazepoxide in a conditioned place preference test [9]. However, although several studies have concluded that there is no stressful impact induced by oral gavage on rodents, they still represent the minority. Wide-spanning literature analysis of various laboratory procedures performed on rodents reveals that orogastric gavage, among other practices, has a significant effect on stress-related responses such as increased heart rate, blood pressure, and corticosterone concentrations [10]. These orogastric gavage studies span from rodents given dry and aqueous gavage to those that are administered gavage either while awake or under anesthesia. An objectively less stressful alternative to orogastric gavage to administer a fecal matter transplant (FMT) is the use of pills, which the subject rodents can simply swallow. However, a switch to pill consumption as a method of FMT has not been popular due to the chemical absorption of the pill as it descends through the GI tract.

While rodent models are the most popular model for studying the microbiome currently, researchers continue to develop alternate models to research this area. One organism that has made a recent introduction into research regarding microbiome manipulation is that of the *Drosophila melongaster (D. melanogaster)*, commonly known as the fruit fly (Figure 1). One of the major reasons that the *D. melanogaster* fly has emerged as a model for manipulation is its ability to create and sustain a thriving environment for microbial species. This practice is known as nutritional mutualism, and researchers can use this ability to harvest bacterial contents that originate directly from the *D. melanogaster* microbiome [11]. While the ability of the fruit fly to readily provide highly concentrated samples of microbiota to researchers makes them a useful model, the absence of any resident commensal strains makes it difficult to draw larger conclusions on host–microbe interactions independent of nutritional influences [12].

Along with the fruit fly and rodent, the zebrafish (*Danio rerio*) has emerged as a key model organism in the study of the microbiome due to its substantial physiological and genetic similarities with humans [13]. In particular, zebrafish possess innate and adaptive immune systems similar to our own. Thus, as microbiome research continues to establish a connection between gut health and the nervous system, the zebrafish model has grown in popularity. The ease with which germ-free zebrafish can be produced in addition to their similar nervous system physiology make them a powerful model in the study of the microbiome (Figure 1). However, the zebrafish model possesses several limitations, primarily due to the aquatic nature of the organism. Zebrafish acquire microbiota solely from aquatic habitats, maintain a temperature of 28 °C at all times, and lack mammary glands and lungs. By contrast, humans derive their microbiome primarily from terrestrial origins, maintain a body temperature nearly ten degrees higher, and could not function without organs such as the lungs. These discrepancies between human and zebrafish physiology greatly limit the direct application of zebrafish studies to clinical research [14].

Thus, current methodology for microbiome research is still lacking. Gnotobiotic animals and heavy antibiotic regimens each yield significant problems in the efficacy of the animal model. To use a subject animal that would begin the study with altered brain function, cytokine expression, and physiology would cast doubt on any conclusions as they relate to humans or even a typical rodent model. Similarly, oral gavage has been shown to be inefficient at best and dangerous at worst in current models. Clearly, there remains a need for a new methodology for microbiome research, one that allows for easy, repeated access to the subject animals’ microbiota.

## 3. The Invisible Manipulator: How the Microbiome Influences Human Physiology

The vast and diverse nature of the microbiome plays many significant roles in human development (Figure 1). Current research shows select microbiota are present in humans as early as at birth. As the newborn is pushed through the vaginal opening, a direct pathway for intergenerational microbial transmission is formed following the rupturing of the chorioamniotic membrane [15]. This exchange provides a foundation for the microbiota to grow over the course of early childhood, with a powerful upsurge in microbiome development at a young age. 

Research has also shown breastfeeding to be a natural pathway for intergenerational sharing of microbiota. A longitudinal study based on gut microbiome development in early childhood defined distinct phases of microbiota progression, which are significantly influenced by breastmilk reception. The microbiome profiles determined by 16S rRNA sequencing of infant stool samples revealed that breastfeeding contributed to significant increases of *Bifidobacterium* species during the first 14 months of life. By an average of 31 months of age, the microbiome reaches a stable state of bacterial diversity [16]. As the field has expanded, research of the early microbiome has revealed evolutionary links between the microbiome and development of human organ systems, especially that of the nervous, circulatory, and gastrointestinal systems. 

Composed of the brain and spinal cord, the central nervous system represents the central compartment of our sensory input, cognition, and motor output. A study comparing the behavioral patterns between GF and conventional mice displayed several instances for the necessity of a microbiota in the development of the nervous system. Results from an elevated maze test suggested that a lack of a microbiome negatively effects cognitive development, as GF mice spent significantly more time in the maze than specific pathogen free (SPF) mice. Additionally, the GF mice displayed an altered expression of motor control and anxiety-like behavior, which is involved in second messenger pathways and synaptic long-term potentiation. Components of gut microbiota appear to regulate the proteins PSD-95 and synaptophysin, which are located in the striatum. These proteins are necessary for synaptogenesis, which is the formation of synapses between neurons within the central nervous system. The results of this study suggest a powerful mechanism for microbial influence on proper brain development [17].

Additionally, the microbiome has a significant influence on the function of glial cells within the nervous system. Glial cells are foundational within the central nervous system, as they form the myelin sheath necessary to protect neurons and allow them to project neural impulses efficiently throughout the body. Profiling microglia from the brains of animals raised under GF conditions revealed significant downregulation of genes necessary for cell activation. Additionally, diminished microglia cell development of GF mice was observed in comparison with SPF mice with normal gut microbiota, emphasizing the necessity of a microbiome in nervous system development [18]. The mechanism of glial cell function plays an important role in modulating spatial and temporal responses to stimuli. A study examining the microglial expression between GF and conventionally raised mice showed that GF mice had a severely diminished response to lipopolysaccharide-induced macrophage activation. Under GF conditions, mice are unable to exhibit functions of glial cells, which is detrimental to the development of myelinating cells throughout the central nervous system [19].

Critical in maintaining an organism’s homeostasis after development of the nervous system is the blood–brain barrier, which describes the selectively permeable nature of blood vessel exchange with the central nervous system. Evidence from studies comparing adult GF mice and conventional mice shows that the gut microbiota also plays a role in the modulation of the blood–brain barrier (BBB). GF mice expressed an increase in blood–brain barrier permeability compared to conventional mice, suggesting that the BBB does not strictly regulate the entry of compounds in the absence of a gut microbiome [19]. Qualitative analysis of mouse embryos from both pathogen-free and GF mothers presented evidence for how the maternal gut microbiota influences the blood–brain barrier of offspring. Antibody infrared-labeled immunoglobin G2b (IgG2b) was administered to the mothers during pregnancy, and the presence of the antibody was assessed in their fetal offspring in order to determine the integrity of the prenatal BBB. Results showed that the IgG2b antibody in GF mothers penetrated to offspring, suggesting a lack of tight junctions in the BBB. Additionally, the expression of occludin, a tight junction protein that helps regulate this epithelial barrier, was significantly lower in the fetal offspring of GF mice [20].

The blood–brain barrier makes up just a small fraction of the blood vessels responsible for transport of nutrients throughout the body. The circulatory system comprises the entirety of these networks, including blood, blood vessels, and the heart itself. Previous research of the gut microbiota has revealed distinct microbial influences on angiogenesis, the development of the blood vessels that comprise our circulatory system. A 2002 study examined the post-natal capillary development of several GF mice using cryosections for Syto61-positive nuclei. Adult GF mice from this study were shown to have arrested growth of capillary networks, suggesting the necessity for a gut microbiome in capillary development. Researchers employed a unique approach to this discovery and were able to induce earlier development of capillary networks in GF mice by inoculation of the gut microbe *Bacteroides thetaiotaomicron* from the cecums of conventionally raised mice. The mechanism linked to this occurrence of angiogenesis is associated with Paneth cells in the intestinal epithelium, which secrete a variety of proteins for immunity and inflammatory control of the mucosa. The GF mice used in this study were determined to be deficient in their overall composition of Paneth cells, so researchers employed an additional comparison of capillary networks using a set of GF mice with intact Paneth cells. Although both sets of mice lacked a microbiome, those lacking Paneth cells had significantly less developed capillary networks, suggesting the importance of Paneth cells during angiogenesis [21]. 

A nutritional study on GF mice revealed that the gut microbiota is necessary in order to regulate the activity of Paneth cells. A group of C57BL/6 mice received an antibiotic cocktail, after which intestinal contents were transferred to GF mice via oral gavage. The antibiotic treatment appeared to decrease lysozyme proteins in the C57BL/6 mice; however, the effects were much more severe in the GF mice. Without a gut microbiome, the GF mice not only had lowered lysozyme levels but also severe atrophy of the small intestine, as well as reduced interleukin mRNA levels. This suggests that the immune defense activities of the Paneth cells were inhibited in the absence of a gut microbiome [22]. This causal link in the function of the intestinal Paneth cells is a pathway in which the gut microbiota plays a vital role for circulatory system development.

At the very focal point of microbiome research is the gastrointestinal (GI) system, containing the organs necessary for digestion. Gut bacteria are beginning to be understood as a prominent link in how this change in the gut microbiome contributes to further development of the GI system itself. Intestinal stem cells of the epithelium are able to grow and differentiate among tissues of the GI system throughout life. Analysis of the *D. melanogaster* fly shows evidence of microbiota-mediated cell proliferation [23]. Acetic acid production is an inherent mediator in the early stages of gut microbiome development. Acetobacter pomorum, a common species of gut microbiota within the *D. melanogaster*, is an acetic acid bacteria with PQQ-ADH activity, which modulates insulin/insulin-like growth factor signaling (IIS). Insulin growth factors such as IIS have a role in GI development through the proliferation and differentiation of intestinal stem cells. Supplementing the simple sugar diet of wild-type, pomorum-monoassociated, and P3G5-monoassociated flies revealed how acetic acid induces IIS activation in P3G5-monoassociated larvae. The activation of IIS was confirmed by examining the membrane targeting of the pleckstrin homology–green fluorescent protein (PH-GFP) [24]. Evidence present in other species link these inherent gut microbes to the development of the GI system. Specialized cells responsible for the growth of the epithelium utilize G-protein-coupled receptors. An additional study on the microbiota-mediated mechanism for stem cell proliferation used mice fed lactic-acid-producing bacteria (LAB), including aforementioned *Bifidobacterium*. Results derived from in situ hybridization staining showed that the lactic acid produced by these bacteria targeted specific G-protein-coupled receptors through Wnt-3/beta-catenin signaling pathways. The targeting of receptor 81 was found to mediate stem cell proliferation [25].

Continued research on intestinal stem cells through a study that housed both GF and SPF mice at weaning age introduces more ways in which the gut microbiota drives GI tissue development. Variable regrowth of intestinal stem cells following radiation-reduced injury revealed the necessity for a microbiome to promote growth of the epithelium. Results revealed that the erythroid differentiation regulator-1 (Erdr1) gene was only induced in the SPF mice, not GF mice. This link was attributed to the Erdr1 gene. Although evidence does not point to a specific interaction between elements within the microbiome and the Erdr1 gene, the presence of a microbiome in itself is necessary for organoid growth within the intestines [26]. The aforementioned mechanisms define the foundational necessity for a microbiome in GI development. Research on the GF rat as far back as 1984 reveals that the cecum can expand up to five times its size in the absence of a microbiome. Alongside the muscle atrophy that occurs without the bacteria necessary for growth, this junction between the small and large intestine experiences a disturbance in solute-water resorption that causes a severe increase in osmotic pressure. The discovery of this abdominal distension in GF rats presents an example of the physical manifestations that can occur due to abnormal GI system development [27].

## 4. What Happens When Microbiome Balance Is Disrupted? Consequences of Dysbiosis

From the previous sections of this review, it is clear that the microbiome represents a crucial facet of early development in humans. However, the rapid expansion of the field of microbiome research continues to shed light on the microbiome as a contributing facet of the etiology and presentation of a wide array of diseases as well. The human microbiome, like any other organ system, does not operate independently. An imbalance in the microbiome, a condition known as gut dysbiosis, can lead to significant downstream effects on the body. However, understanding this connection between dysbiosis and disease may allow researchers to better understand the etiology of some of the most prevalent diseases in society, and develop new avenues for their treatment.

As mentioned previously, one of the chief mechanisms for the interplay between the microbiome and disease is the gut–brain–microbiome axis. Though most studies are still in preclinical stages, evidence continues to point towards a circular communication loop along this axis, as the brain responds to signaling from the gut microbiome in order to modify gut motility and permeability to in turn affect the functionality of the microbiome [28]. It is along this axis that researchers are now beginning to find a wide range of interactions between the gut, brain, microbiome, and disease. 

One of the most recent connections made by researchers is that between the gut–brain–microbiome axis and several neurodegenerative diseases, such as Parkinson’s and Alzheimer’s disease (AD). Current research supports the significance of amyloid- β peptides as the main constituent of amyloid plaques found in the brains of those with AD [29]. Similarly, Parkinson’s disease dementia emerges due to the abnormal aggregation of Lewy bodies, another neuroprotein. Research continues to show a link between the microbiome, these neurodegenerative diseases connected by the gut–brain axis, and, in particular, the vagal nerve (cranial nerve 10), with a novel 2014 study showing the ability of the neuroprotein alpha-synuclein to travel along the vagal nerve in vivo [30]. While research is still ongoing, another 2014 study identified the significance of gut bacteria in the production of amyloids like those seen in AD which, along with the aforementioned vagal nerve study, highlights a key possible source of these harmful amyloids [31]. In addition, Marizzoni et al. reported that gut microbiota-related products, including lipopolisaccharides and short-chain fatty acids, and systemic inflammation are associated with the presence of brain amyloidosis in older human subjects. This association appears to be a consequence of endothelial dysfunction [32]. Thus, continued research may uncover a clinically significant connection between the gut microbiome and these neurodegenerative diseases. 

Research has also shown a link between gut dysbiosis and the pathogenesis of immunodeficiency viruses such as Human Immunodeficiency Virus (HIV) and Simian Immunodeficiency Virus (SIV). Loss of beneficial bacteria such as lactobacillus and the increased occurrence of harmful bacteria such as acinetobacter have been previously linked with the pathogenesis of HIV, resulting in digestive issues and inflammation, respectively [33,34]. A 2016 study examining the effect of fecal microbiota transplantation (FMT) with SIV positive test subjects found that the FMT induced a positive response in the animals, improving their health, and reducing CD4+ T cell activation, and therefore inflammation, in the gut [35]. 

As one would expect, gut dysbiosis has also been shown to have a dramatic effect on gut health. As published in 2008 by the American Gastroenterologist Association’s journal, Crohn’s disease, ulcerative colitis (UC), and pouchitis are all the result of pathogenic immune response following the antigenic stimulation by microbiota in the gut, due to mucosal barrier defects, or immunoregulation [36]. Not only is dysbiosis a frequent factor in the etiology of these diseases, but also in their severity. A 2007 study found that Crohn’s disease and Inflammatory Bowel Disease patients with abnormal microbiota received surgical intervention for their disease at a significantly younger age than those with healthy microbiomes (8.1 years; *p* = 0.002) and showed a significant correlation between abnormal microbiota and abscess occurrence [37]. 

Yet, perhaps the most researched link between dysbiosis, particularly of the gut, and disease, is the Clostridioides difficile Infection (CDI). CDI is an opportunistic infection caused by the *Clostridioides difficile* bacteria, and it represents one of the most prevalent causes of infectious colitis in hospitalized patients [38]. The infection, which most often arises following disruption of the gut microbiome due to the use of antibiotics, is one of the most frequent causes of disease and death in adult hospitalized patients, with a 2018 panel convened by the Infectious Diseases Society of America (IDSA) and Society for Healthcare Epidemiology of America (SHEA) citing it as “the most commonly identified cause of healthcare-associated infection in adults in the United States” [39]. 

With CDI occurring in patients with disrupted intestinal microbiota, it follows that a healthy gut microbiome can prevent the colonization of the *Clostridioides difficile* bacteria. This mechanism, known as “colonization resistance”, suggests that established and healthy microbiota will outcompete the pathogen, stimulate the development of protective mucosal barriers, produce inhibitory substances, and stimulate the host immune system [40]. Thus, recent research has sought out new ways of establishing colonization resistance in CDI patients to prevent reinfection following antibiotics. However, it is important to note that it is still largely unknown whether it is the lack of certain bacteria that cause disease or whether it is the disease that causes the lack of bacteria.

## 5. How Do We Restore a Healthy Microbiome? Current Treatments for Dysbiosis

The imbalance of bacterial species within the human gut microbiota proves to be a significant commonality among the onset of various debilitating conditions (Figure 1). As these correlations between gut dysbiosis and disease become prevalent in current research of the microbiome, several measures have been taken in expanding such discoveries through the application of medical supplementation and treatments. Likely for all of human history, the microbiome has influenced the physiology and disease of humans, yet only recently have researchers begun to understand how they might alter the physiology of the microbiome to improve life. Current medical practices have proven to reduce bacterial disruption within the colon that manifests into the aforementioned infection known as *Clostridium Difficile* (*C. difficile*). The therapeutic treatment for *C. difficile* is implemented through FMT, which intends to introduce healthy bacteria in order to favorably restore the gut bacteria of infected individuals. 

A case study following a recipient of FMT for recurrent *C. difficile* confers evidence for the mechanistic influence of gut bacteria following use of this form of bacteriotherapy. Using 16S rRNA sequencing of colonic bacteria, the bacterial composition showed a significant increase in strains of Bacteroidetes species after treatment, which was deficient prior to treatment. Following FMT, the patient’s microbiome composition closely resembled that of the donor, suggesting that healthy bacteria are able to effectively integrate with the recipient [41]. An additional study that analyzed the microbiome composition of patients following FMT showed a similar trend in the resulting bacterial restoration of the gut. Using 16S rRNA sequencing, the two patients previously deficient in Bacteroidetes species were shown to have been restored following FMT. The composition in Bacteroidetes abundance was also found to be very similar to that of the donor sample. The results collected from this microbial analysis suggested that the effectiveness in FMT is based upon the interactional mechanism of engraftment of the donor microbiota, in that the healthy bacteria begin to repopulate within the recipient [42]. 

FMT is carried out through the administration of a donor’s fecal microbiota via ingestion of oral capsules, transfer by nasoduodenal tube, or colonoscopy. In current practices, FMT is most commonly performed through colonoscopy to minimize adverse effects associated with oral ingestion. Ingestion of oral capsules has shown to successfully relieve *C. difficile* symptoms as outlined by a cohort study of 20 patients in which a 90% clinical resolution of diarrhea was observed. Although this study supports the efficacy of oral ingestion, aspiration of fecal matter is a concern with this method of treatment [43]. Administering FMT through a nasoduodenal tube can pose the same health concerns, as one case report attributed fatal aspiration pneumonia to this method of treatment. A complication in this patient’s procedure caused regurgitation of the feculent liquid, which later resulted in sepsis that was attributed to a case of pneumonia [44]. With the overall goal of minimizing complications and ensuring effective transplantation, several regulatory measures are taken for administration through colonoscopy. In addition to taking note of patients’ medical histories, a donor stool specimen is provided by a healthy individual who passed an extensive pathogenic blood test [45].

New methods of material management have helped simplify FMT as it has integrated into medical practice as a standard means of *C. difficile* treatment. As a way to overcome the difficulty in searching for reliable donors to be readily available for the FMT procedure, clinical practice from the University of Minnesota has revealed that donor fecal material can be frozen and ready for use when needed. Treatment of 43 patients using previously frozen donor fecal material showed that this simplification of material storage was just as effective as previous FMT applications [46]. An additional study utilizing frozen donor fecal bacteria in comparison to fresh fecal bacteria between two patients revealed similar effectiveness in *C. difficile* treatment. This outcome was confirmed through 16s rRNA sequencing of the patients’ gut microbiota, revealing similar compositions with an abundance of the Bacteroidetes phylum post FMT [42]. As a way to further enhance the mobility and storage of donor fecal samples, lyophilization of FMT preparations resulted in similarly effective treatment. Lyophilization is the freeze-drying of a stool sample via sublimation to create a powder-like product. Results from a randomized clinical trial comparing FMT administered by colonoscopy among fresh, frozen, and lyophilized samples showed a slightly lower cure rate of *C. difficile* symptoms using lyophilized donor fecal matter. Although fresh and frozen donor samples were determined to be statistically more effective, a majority of treated individuals still benefitted from treatment using lyophilized product [47].

The efficacy of FMT is dependent on the viability of the microbes that are able to be transplanted into the recipient. In an attempt to investigate the conditions that affect the survivability of bacterial communities within fecal samples, researchers utilized polymerase chain reaction (PCR) and 16sRNA gene sequencing to analyze eight stool samples from healthy donors. While the freezing of samples did reduce the viability of their overall bacteria, the samples still maintained a proportional composition of microbial species. A more significant factor contributing to the reduced survivability of samples was observed to be due to atmospheric oxygen. Exposing the samples to ambient air resulted in a 12-fold reduction in several butyrogenic species [48]. This correlation between air exposure and the loss of butyrogenic species provided a potential explanation for the ineffectiveness of some FMT applications. The depletion of butyrogenic bacteria in the gut microbiome is a commonality among individuals with *C. difficile* infections, as evident from a clinical survey using genomic analysis of the stool samples of 39 affected individuals in comparison to healthy controls [49]. Without the bacterial species required to sustain a healthy microbiome, the donor samples devoid of butyrogenic species would be unable to provide effective treatment using FMT.

Additional methods of manipulating the gut microbiome as an attempt to treat gut dysbiosis have come to light through the understanding of how certain foods impact our digestion. Present in a large variety of fruits and vegetables, prebiotics are substrates that help regulate and fortify beneficial bacteria within the host. These food ingredients were first encountered as a factor within human milk that effectively increased the presence of bifidobacteria within infant microbiomes. Through this analysis of infant stool samples from a study dating back to 1954, researchers were unsure of what to attribute to this mechanism of bacterial growth [50]. Years later, this interaction among microbial communities revealed these compounds were comprised of oligosaccharides and glycan, which are now understood to be a series of indigestible nutrients that provide fuel for bacteria to enhance growth. The growth of beneficial gut microbials was specifically that of bifidobacteria and lactic acid bacteria [51]. Additional compounds regarded as prebiotics include fructo-oligosaccharides, galacto-oligosaccharides, inulin, and lactulose. Although these compounds are naturally found as fermentable fiber within food, isolation techniques including that of ultrasound treatment have allowed prebiotics to be taken as oral supplements [52]. 

Since the discovery of its beneficial potential, several studies have outlined the efficacy of prebiotic ingestion on the gut microbiome. Utilizing a powder form of galacto-oligosaccharide, a type of prebiotic fiber, a cohort study of elderly individuals examined the presence of bifidobacteria within the gut microbiome. Subjects consumed 5.5 g of the powder mixed in with water each day at about the same time, for a total of 10 weeks. During this study, stool samples were collected and specific segments of bacterial nucleic acids isolated via in situ hybridization. Results collected over this period suggested that the consumption of galacto-oligosaccharides significantly increased levels of bacteroides and bifidobacteria [53]. Additionally, the prebiotic inulin follows a similar trend, resulting in the increase of hosts’ bifidobacteria composition. Volunteers in this study ingested 10 g of inulin daily over a 16-day period. PCR analysis of the volunteers’ stool samples revealed that *Bifidobacterium adolescentis* showed a strong increase as a response to inulin consumption [54]. Alongside the recomposition of bacterial species within the gut microbiome, studies also shed light on the positive role that functional prebiotics can play in the alleviation of symptoms associated with irritable bowel syndrome (IBS). In a 12-week clinical trial, 44 patients with IBS received 7 g doses of prebiotic trans-galactooligosaccharide. Stool analysis of the treatment group revealed that the prebiotic increased numbers of gram-positive bacteria, while effectively alleviating IBS symptoms [49].

Increasingly prevalent as a food additive and supplemental approach to the treatment of gut dysbiosis, probiotics are live bacteria and yeasts designed to beneficially integrate with host bacteria. Although the efficacy of probiotic supplementation is often contested as to the precise effects it has on the host microbiome, a placebo-controlled trial on probiotic-mediated colonization of bacterial species revealed that the effects are often predictive of those already present in the host. Researchers investigated the comprehensive microbial composition found along the GI tract using colonoscopy and endoscopy procedures. Samples were collected both prior and three weeks after bi-daily probiotic supplementation over a 4-week period. Species-specific pPCR quantification revealed that significant colonization only occurred in individuals with lower levels of *Bifidobacterium breve* and *Bifidobacterium infantis* prior to supplementation. This evidence suggests that probiotics may only be effective when the host is deficient in a certain bacterial species [55]. It should also be noted that probiotic supplementation could be disruptive to the recovery timeline of gut microbial composition following antibiotic treatment. A longitudinal study comparing the reconfiguration of bacterial composition in individuals treated with autologous fecal microbiome transplantation (aFMT), probiotic supplementation, or spontaneous recovery after using antibiotics revealed significantly altered outcomes. Volunteers in this study received standard dosages of ciprofloxacin and metronidazole for seven days. Biopsies were performed via colonoscopy and endoscopy both prior to and three weeks after antibiotic treatment. QPCR-based quantification revealed that dysbiosis remained present in those receiving bi-daily 11-strain probiotic supplements, whereas microbial compositions were restored to pre-treatment levels in aFMT and spontaneous recovery groups after the 28-day intervention period. These results highlight potentially unfavorable outcomes associated with probiotic usage in contrast to previously discussed FMT treatments [56].

Nonetheless, several studies outline the efficacy of probiotic use in alleviating symptoms associated with manifestations of gut dysbiosis, including pouchitis and ulcerative colitis. Ulcerative colitis (UC) is an inflammatory disease of the GI tract. Evidence from a randomized trial of patients with UC shows how a probiotic supplement was able to effectively relieve symptoms and induce remission. Patients were treated with a probiotic preparation consisting of lyophilized bacteria of lactobacilli, bifidobacteria, and Streptococcus thermophilus. This probiotic mixture is known as VSL#3, and is used as treatment for several disorders relating to gut dysbiosis. Compared to the placebo group, the treatment group experienced 20% higher rates of remission and symptom improvement outlined by the ulcerative colitis disease activity index (UCDAI) [57]. Associated with an inflammation of the ileal pouch, pouchitis occurs in people who have received J-pouch surgery as a treatment for ulcerative colitis. This inflammation results in unpleasant symptoms of increased stool urgency and frequency, as well as abdominal cramping. A randomized controlled trial assigned a daily 6 g probiotic supplement of 5 × 10^11^ per gram VSL#3 to patients experiencing a history of relapsing pouchitis. Nine months of treatment resulted in significantly fewer relapses. At the end of treatment, however, all of the patients experienced a relapse, with fecal lactobacillus and bifidobacteria [58].

Probiotics must be able to survive the harshly acidic environment of the GI tract in order to relay benefits to the host microbiome. Acid tolerance is one of the most desirable characteristics considered in selecting bacterial species for probiotic supplements. This has greatly popularized lactobacillus for its inherent ability to occupy itself within the epithelium of the small intestine, as determined from a longitudinal study of fecal PCR analysis [59]. An in vitro assessment of dairy Propionibacterium also showed promising efficiency as a probiotic, proving viability in a pH of 2.0 [60]. Selection for bacteria that can tolerate the harsh acidity of the stomach (normal pH 1.5–3.5) and slight alkalinity of bile (pH 7–8) does not guarantee that a large enough concentration of live cells will be able to reach the small intestine for absorption. Results from an in vitro analysis of gastric survival rates for several commercial probiotics revealed that even strains expected to withstand enteric protection were not viable when taken on an empty stomach [61]. To combat this, probiotics administration in food requires a form of protection during ingestion. This technology has been introduced through the use of lipid coating for bacterial microencapsulation. An in vitro study of probiotic gastrointestinal delivery observed that non-encapsulated cells were destroyed, whereas microencapsulated probiotics strains were delivered while keeping their functional properties intact [62]. An experimental study on the encapsulation of Bifidobacterium and *Lactobacillus* subjected several probiotic cultures to varying pH levels and temperatures in order to determine the efficiency of encapsulation with hydrogel beads. The encapsulation efficiency (%) was determined by the following equation:$$\frac{\mathrm{Cell}\;\mathrm{count}\;\left(\mathrm{CFU}/\mathrm{mL}\right)\;\mathrm{after}\;\mathrm{disintegration}\;\mathrm{of}\;\mathrm{the}\;\mathrm{hyrdrogel}\;\mathrm{beads}}{\mathrm{Initial}\;\mathrm{loading}\;\mathrm{of}\;\mathrm{the}\;\mathrm{cells}\;\left(\frac{\mathrm{CFU}}{\mathrm{mL}}\right)\mathrm{in}\;\mathrm{hydrogel}\;\mathrm{beads}}$$

CFU represents bacterial colony forming units. Probiotics with a higher encapsulation efficiency were able to best withstand pH levels as low as 2, with survival rates at around 6 log CFU mL^−1^. High encapsulation efficiencies also allowed bacteria to better survive in high temperature conditions. A temperature of 50 °C was determined to be the most ideal for probiotic survival, which was around 9 log CFU mL^−1^ [63]. Probiotic transport can also be aided by ingesting bacteria alongside food products. An in vitro study of oral probiotic ingestion successfully mimicked the protection provided by encapsulation of bacterial species by preparing products in a buttermilk base. After providing a dose of at least 10^6^ CFU each day, the strains survived for up to two weeks in the GI tract following ingestion [64]. With the current use of medical treatment and supplementation methods, the gut microbiome is becoming a desirable area of study for future medical intervention. Although the various therapeutics outlined for gut dysbiosis play a role in temporary relief, continued research aims to discover options that can provide lasting enhancement for the human gut microbiome.

## 6. Advancing Modern Medicine through the Microbiome

As previously discussed, FMT appears to be one of the clear routes forward for the treatment of gut dysbiosis. Studies continue to show its efficacy both in the treatment of CD currently, and the possible clinical treatment of irritable bowel disease (IBD) in the future. One 2015 study showed that, in a study of nine pediatric patients, seven achieved remission within two weeks of FMT, and five were in remission at six and twelve weeks after FMT without any additional medical therapy [65]. A similar study among adults from Wei et al. found significant improvements in quality of life among the study’s 14 IBD patients [66]. 

However, FMT is still a novel procedure, and its long-term effects are not yet well-known. Additionally, the methods by which donors are screened is still in a continuous development process. Factors such as gender, weight, and even age can greatly affect one’s microbiome. One 2017 study observed the outcomes of FMT from samples provided by 28 donors, eight of whom were over 60. In this study, elderly fecal samples were shown to be just as efficacious as those from their younger counterparts. However, seven out of eight recipients of fecal samples from elderly donors were above the age of 60, and four out of eight of the recipients were spouses of their donors. The study also showed that older groups had undergone compositional alterations in fecal microbiota, suggesting that perhaps their age may still be a factor in FMT donor screening [67]. 

Though the aforementioned studies did not report any adverse side effects of the treatment, FMT is still a relatively new procedure. There are trillions of bacteria in our gut, and future research must focus on which of these are beneficial and which are harmful in order to improve the efficacy of treatments such as FMT. However, this unique microbial fingerprint presents a novel pathway for scientists to develop the clinical applications of this research. The wide array of diseases shown to be connected to the microbiome in their etiology, ranging from neurological to gastrointestinal in nature, highlight the broad range of applications the results of this research may have. Going forward, it is likely that researchers will begin to marry the two popular concepts of FMT and probiotic supplementation to create targeted microbiota treatments, uniquely composed based on the patient’s disease, gender, age, and/or any other unique factor. Though a large gap still remains between laboratory research and clinical applications, there is no doubt that the microbiome will play a substantial role in the future of modern medicine. 

## Figures and Tables

**Figure 1 nutrients-13-00074-f001:**
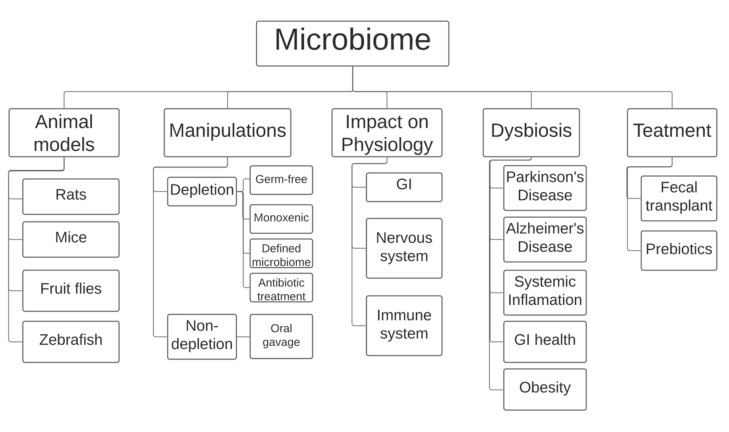
Currently, the field of microbiome research employs rodents (mice and rats), the fly Drosophila melanogaster, and the fish zebrafish as animal models. Most manipulations involve either complete/partial depletion of the microbiome using specific rodent models or microbiome transplants using oral gavage. Research has shown that the microbiome is heavily involved in improper development of the gastrointestinal tract and the nervous and immune systems. A link has also been made between imbalances in the gut microbiome, also known as dysbiosis, and certain diseases, including Parkinson’s and Alzheimer’s disease and obesity, among others. Gut dysbiosis is currently treated in humans using prebiotics and/or fecal transplants. GI: gastro-intestinal tract.

## Data Availability

All data obtained from published papers. Source PubMed.

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
