# Peer review of "Managing the Microbiome: How the Gut Influences Development and Disease"

_nutrients, 2020, doi:10.3390/nu13010074_

Round 1
Reviewer 1 Report
Weinstein et al present a very interesting review of the relationship between the microbiome and host development and disease progression. This manuscript highlights the role the microbiome plays in immune, nervous, and GI development in the host. Also, the authors summarize current research on the gut-brain axis and the microbiome's relationship with Alzheimer's disease and Parkinson's disease. I think this is an interesting review but needs some substantial structural changes (especially section 5).
Minor changes
- Page 1 line 29: 'microbiome' does not need to be capitalized.
- Page 2 line 51: ‘Mature’ Is the wrong word here. You are correct that the microbiome diversifies rapidly until about the age of 3, but it continues to evolve until the age of 40 and then will stabilize. (see Age- and Sex-Dependent Patterns of Gut Microbial Diversity in Human Adults by Jacobo de la Cuesta-Zuluaga )
- Page 2 line 76: You need to define LPS. Also, LPS stimulation seems out of place in this paragraph all about the nervous system. I suggest removing the part about LPS or further explain the relationship.
- Page 3 line 132: It should be ‘gut microbes’ not ‘gut microbials’
- Page 4 line 195: This should be inflammatory (not irritable) bowel disease
- Page 5 line 223: You need to define FMT
- Page 6 line 294: I believe ‘probiotics’ should actually be ‘prebiotics’
- Page 7 line 304: I think it would be helpful to the reader to mention that galacto-oligosaccharides are a type of fiber
- Page 7 line 324: You should mention that pouchitis only occur in people who have the J pouch surgery (also a UC treatment).
- Page 10 line 489: Why do you switch reference style?
- If you are going to use GF to stand for germ-free I think this needs to be consistent throughout the paper. As it stands now there are many instances of both GF and germ-free being used which may be confusing to readers.
Major changes
- At the end of the third section, I think it is important to mention that it is still largely unknown the direction of some of these effects. Does the lack of certain bacteria cause the disease, or does the disease cause the lack of bacteria?
- In your probiotic section (in part four) of this manuscript, I think you can add in some details from these two papers. First is Personalized Gut Mucosal Colonization Resistance to Empiric Probiotics Is Associated with Unique Host and Microbiome Features by Zmora et al, which talks about how most probiotics only really ‘work’ if the patient is deficient in that bacteria. Second is Post-Antibiotic Gut Mucosal Microbiome Reconstitution Is Impaired by Probiotics and Improved by Autologous FMT by Suez et al, which suggests that probiotics might hinder the reconstruction of the microbiome after antibiotic treatment.
- Page 8 line 365: I think you need to restructure this section and work on a better transition. As it stands now you go from probiotics and re-establishing the microbiome to animal models which is a little jarring. Also, you have talked about germ-free models extensively above but are just now going into detail about what they are. You might want to move this section to the beginning of the paper.
- Page 10 line 495: I think this is another jarring transition going from mice to flys then back to mice. I would move this to the end of the section and condense this paragraph into a sentence or two.
- Your fifth section needs to be heavily cut down. While model organisms are important, the scope of this paper is how the microbiome is involved with development and disease, not model organisms. Personally, I would try to condense this section into 2 or 3 paragraphs and have that be the second section of the paper (right after your introduction). To do this, I would cut (or substantially edit down) the sections on how to develop GF mice. I would also edit down the paragraph on drosophila and about oral gavage (important to mention but no need to spend 6-7 paragraphs on it)
- This manuscript would benefit from at least one figure to break up the text a bit.
Overall I think this is a well-written review that provides new information for the field. My largest concern is the animal model section (as mentioned above). Personally, I feel like this section does not fit with the rest of the manuscript and should be cut down to 2-3 paragraphs. I also think this cut-down version should be moved to the beginning of the paper (right before section 2). I also think this paper would benefit from a figure or two which would help break up the text.
Author Response
Cover letter
Journal: Nutrients
Manuscript ID: nutrients-1040190
Title of paper: Managing the Microbiome: How the Gut Influences Development and Disease
Authors: Noah Weinstein, Brandon Garten, Jonathan Vainer, Dulce M. Minaya, Krzysztof Czaja *
Ms. Sarah Zhao,
We appreciate the time and efforts by the Editor and Referees in reviewing this manuscript. We have addressed all issues indicated in the review report and believe that the revised version meets the journal’s publication requirements.
Response to comments from Reviewer # 1:
Weinstein et al present a very interesting review of the relationship between the microbiome and host development and disease progression. This manuscript highlights the role the microbiome plays in immune, nervous, and GI development in the host. Also, the authors summarize current research on the gut-brain axis and the microbiome's relationship with Alzheimer's disease and Parkinson's disease. I think this is an interesting review but needs some substantial structural changes (especially section 5).
Minor changes
- Page 1 line 29: 'microbiome' does not need to be capitalized.
- Response: This has been corrected as suggested.
- Page 2 line 51: ‘Mature’ Is the wrong word here. You are correct that the microbiome diversifies rapidly until about the age of 3, but it continues to evolve until the age of 40 and then will stabilize. (see Age- and Sex-Dependent Patterns of Gut Microbial Diversity in Human Adults by Jacobo de la Cuesta-Zuluaga)
- Response: You are correct. Thus, the word mature has been deleted. This is now Page 4, line 135.
- Page 2 line 76: You need to define LPS. Also, LPS stimulation seems out of place in this paragraph all about the nervous system. I suggest removing the part about LPS or further explain the relationship.
- Response: The section has been revised to read “A study examining the microglial expression between GF and conventionally raised mice showed that GF mice had a severely diminished response to Lipopolysaccharide-induced macrophage activation. Under GF conditions, mice are unable to exhibit functions of glial cells, which is detrimental to the development of myelinating cells throughout the central nervous system”. This is now page 4 line 155.
- Page 3 line 132: It should be ‘gut microbes’ not ‘gut microbials’
- Response: The word microbials has been changed to microbes. This is now page 5 line 210.
- Page 4 line 195: This should be inflammatory (not irritable) bowel disease
- Response: The word irritable has been changed to inflammatory. This is now page 6 line 271.
- Page 5 line 223: You need to define FMT
- Response: Fecal matter transplant (FMT) is defined upon first use (page 3 line 95)
- Page 6 line 294: I believe ‘probiotics’ should actually be ‘prebiotics’
- Response: The word probiotics has been changed to prebiotics. This is now page 8 line 364.
- Page 7 line 304: I think it would be helpful to the reader to mention that galacto-oligosaccharides are a type of fiber
- Response: Thank you for the recommendation. We have modified the sentence to read “Utilizing a powder form of galacto-oligosaccharide, a type of prebiotic fiber, a cohort study of elderly individuals examined the presence of bifidobacteria within the gut microbiome.” This is now page 8 line 369.
- Page 7 line 324: You should mention that pouchitis only occur in people who have the J pouch surgery (also a UC treatment).
- Response: Thank you for the recommendation. We have modified the sentence to read “Associated with an inflammation of the ileal pouch, pouchitis occurs in people who have received J-pouch surgery as a treatment for ulcerative colitis. This inflammation results in unpleasant symptoms of increased stool urgency and frequency, as well as abdominal cramping.” This is now page 8 line 411.
- Page 10 line 489: Why do you switch reference style?
- Response: We have made sure that references throughout the manuscript conform to the same style.
- If you are going to use GF to stand for germ-free I think this needs to be consistent throughout the paper. As it stands now there are many instances of both GF and germ-free being used which may be confusing to readers.
- Response: Germ-free was defined upon first use and GF used throughout the manuscript thereafter.
Major changes
- At the end of the third section, I think it is important to mention that it is still largely unknown the direction of some of these effects. Does the lack of certain bacteria cause the disease, or does the disease cause the lack of bacteria?
- Response: We agree with the recommendation and the following sentence has been added: “However, it is important to note that it is still largely unknown whether it is the lack of certain bacteria that cause disease or it is the disease that causes the lack of bacteria.” This is now the end of the fourth section, page 7 line 288.
- In your probiotic section (in part four) of this manuscript, I think you can add in some details from these two papers. First is Personalized Gut Mucosal Colonization Resistance to Empiric Probiotics Is Associated with Unique Host and Microbiome Features by Zmora et al, which talks about how most probiotics only really ‘work’ if the patient is deficient in that bacteria. Second is Post-Antibiotic Gut Mucosal Microbiome Reconstitution Is Impaired by Probiotics and Improved by Autologous FMT by Suez et al, which suggests that probiotics might hinder the reconstruction of the microbiome after antibiotic treatment.
- Response: We appreciate your recommendation and have added the following details to the start of our discussion on probiotics: “Although the efficacy of probiotic supplementation is often contested as to the precise effects it has on the host microbiome, a placebo-controlled trial on probiotic-mediated colonization of bacterial species revealed that the effects are often predictive of those already present in the host. Researchers investigated the comprehensive microbial composition found along the GI tract using colonoscopy and endoscopy procedures. Samples were collected both prior and 3 weeks after bi-daily probiotic supplementation over a 4-week period. Species-specific pPCR quantification revealed that significant colonization only occurred in individuals with lower levels of Bifidobacterium breve and Bifidobacterium infantis prior to supplementation. This evidence suggests that probiotics may only be effective when the host is deficient in a certain bacterial species [54]. It should also be noted that probiotic supplementation could be disruptive to the recovery timeline of gut microbial composition following antibiotic treatment. A longitudinal study comparing the reconfiguration of bacterial composition in individuals treated with autologous fecal microbiome transplantation (aFMT), probiotic supplementation, or spontaneous recovery after using antibiotics revealed significantly altered outcomes. Volunteers in this study received standard dosages of ciprofloxacin and metronidazole 7 days. Biopsies were performed via colonoscopy and endoscopy both prior to and 3 weeks after antibiotic treatment. QPCR-based quantification revealed that dysbiosis remained present in those receiving bi-daily 11-strain probiotic supplements, whereas microbial compositions were restored to pre-treatment levels in aFMT and spontaneous recovery groups after the 28-day intervention period. These results highlight potentially unfavorable outcomes associated with probiotic usage in contrast to previously discussed FMT treatments [55]. Nonetheless, several studies outline the efficacy of probiotic use in alleviating symptoms associated with manifestations of gut dysbiosis, including pouchitis and ulcerative colitis.” Page 8 line 384. The recommended citations have been included.
- Page 8 line 365: I think you need to restructure this section and work on a better transition. As it stands now you go from probiotics and re-establishing the microbiome to animal models which is a little jarring. Also, you have talked about germ-free models extensively above but are just now going into detail about what they are. You might want to move this section to the beginning of the paper.
- Response: We agree. Per your recommendation, this section was heavily edited, and it is now section two.
- Page 10 line 495: I think this is another jarring transition going from mice to flys then back to mice. I would move this to the end of the section and condense this paragraph into a sentence or two.
- Response: Thank you for pointing this out. This was addressed while restructuring this section.
- Your fifth section needs to be heavily cut down. While model organisms are important, the scope of this paper is how the microbiome is involved with development and disease, not model organisms. Personally, I would try to condense this section into 2 or 3 paragraphs and have that be the second section of the paper (right after your introduction). To do this, I would cut (or substantially edit down) the sections on how to develop GF mice. I would also edit down the paragraph on drosophila and about oral gavage (important to mention but no need to spend 6-7 paragraphs on it)
- Response: We agree with the reviewer that this section was too lengthy. Thus, we have condensed it and moved it to the second section of the manuscript.
- This manuscript would benefit from at least one figure to break up the text a bit.
- Response: Thank you for the recommendation. We have created figure one to guide the reader.
Sincerely,
Dr. Krzysztof Czaja

Reviewer 2 Report
Ms. ID nutrients-1040190: Managing the Microbiome: How the Gut Influences Development and Disease
General comments:
Weinstein et al. have submitted a literature review that stands out by offering the reader a critical look at microbiome studies. Indeed, with convincing examples, the authors summarize the advances that have been made in certain areas such as the blood-brain barrier.
Specific comments:
Lines 169-181: Please consider adding the following reference:
A recent study confirms the correlation, in humans, between an imbalance in the gut microbiota and the development of amyloid plaques in the brain, which are at the origin of the neurodegenerative disorders characteristic of Alzheimer's disease. I suggest to introduce this aspect: gut microbiota-related products and systematic inflammation
- Moira Marizzoni, Annamaria Cattaneo, Peppino Mirabelli, Cristina Festari, Nicola Lopizzo, Valentina Nicolosi, Elisa Mombelli, Monica Mazzelli, Delia Luongo, Daniele Naviglio, Luigi Coppola, Marco Salvatore, Giovanni B. Frisoni. Short-Chain Fatty Acids and Lipopolysaccharide as Mediators Between Gut Dysbiosis and Amyloid Pathology in Alzheimer’s Disease. Journal of Alzheimer's Disease, 2020; 78 (2): 683 DOI: 10.3233/JAD-200306
Lines 459-482: The authors put a lot of emphasis on Drosophila as a model. It would have been appropriate to also mention the zebrafish as a model which possesses similar central nervous system (CNS) morphology and genetics with human for studying host-microbe interactions and the mechanistic way of gut-brain function. Please consider to cite Kamareddine et al. (2020)
Mohanta, L., Das, B. C., & Patri, M. (2020). Microbial communities modulating brain functioning and behaviors in zebrafish: A mechanistic approach. Microbial Pathogenesis, 104251.
Kamareddine, L., Najjar, H., Sohail, M. U., Abdulkader, H., & Al-Asmakh, M. (2020). The Microbiota and Gut-Related Disorders: Insights from Animal Models. Cells, 9(11), 2401.
Minor comments:
Line 50 and throughout the text: Bifidobacterium. Comments: Please refer to
https://jcm.asm.org/content/nomenclature
“Names of all taxa (kingdoms, phyla, classes, orders, families, genera, species, and subspecies) are printed in italics and should be italicized in the manuscript; strain designations and numbers are not”.
Author Response
Cover letter
Journal: Nutrients
Manuscript ID: nutrients-1040190
Title of paper: Managing the Microbiome: How the Gut Influences Development and Disease
Authors: Noah Weinstein, Brandon Garten, Jonathan Vainer, Dulce M. Minaya, Krzysztof Czaja *
Ms. Sarah Zhao,
We appreciate the time and efforts by the Editor and Referees in reviewing this manuscript. We have addressed all issues indicated in the review report and believe that the revised version meets the journal’s publication requirements.
Response to comments from Reviewer # 2:
General comments:
Weinstein et al. have submitted a literature review that stands out by offering the reader a critical look at microbiome studies. Indeed, with convincing examples, the authors summarize the advances that have been made in certain areas such as the blood-brain barrier.
Specific comments:
Lines 169-181: Please consider adding the following reference:
A recent study confirms the correlation, in humans, between an imbalance in the gut microbiota and the development of amyloid plaques in the brain, which are at the origin of the neurodegenerative disorders characteristic of Alzheimer's disease. I suggest to introduce this aspect: gut microbiota-related products and systematic inflammation
- Moira Marizzoni, Annamaria Cattaneo, Peppino Mirabelli, Cristina Festari, Nicola Lopizzo, Valentina Nicolosi, Elisa Mombelli, Monica Mazzelli, Delia Luongo, Daniele Naviglio, Luigi Coppola, Marco Salvatore, Giovanni B. Frisoni. Short-Chain Fatty Acids and Lipopolysaccharide as Mediators Between Gut Dysbiosis and Amyloid Pathology in Alzheimer’s Disease. Journal of Alzheimer's Disease, 2020; 78 (2): 683 DOI: 10.3233/JAD-200306
- Response: We appreciate the recommendation. The following sentence has been added “In addition, Marizzoni et al. reported that gut microbiota related products, including lipopolisaccharides and short-chain fatty acids, and systemic inflammation are associated with the presence of brain amyloidosis in older human subjects. This associating appears to be a consequence of endothelial dysfunction”. Page 6 line 253. The suggested citation was included.
- Lines 459-482: The authors put a lot of emphasis on Drosophila as a model. It would have been appropriate to also mention the zebrafish as a model which possesses similar central nervous system (CNS) morphology and genetics with human for studying host-microbe interactions and the mechanistic way of gut-brain function. Please consider to cite Kamareddine et al. (2020)
Mohanta, L., Das, B. C., & Patri, M. (2020). Microbial communities modulating brain functioning and behaviors in zebrafish: A mechanistic approach. Microbial Pathogenesis, 104251.
Kamareddine, L., Najjar, H., Sohail, M. U., Abdulkader, H., & Al-Asmakh, M. (2020). The Microbiota and Gut-Related Disorders: Insights from Animal Models. Cells, 9(11), 2401.
- Response: We appreciate the recommendation. The following paragraph has been added to page 3 line 105 “Along with the fruit fly and rodent, the Zebrafish (Danio rerio) has emerged as a key model organism in the study of the microbiome due to its substantial physiological and genetic similarities with humans [12]. In particular, Zebrafish possess innate and adaptive immune systems similar to our own. Thus, as microbiome research continues to establish a connection between gut health and the nervous system, the Zebrafish model has grown in popularity. The Zebrafish model poses several limitations, primarily due to the aquatic nature of the organism; Zebrafish acquire microbiota solely from aquatic habitats, maintain a temperature of 28 â—¦C at all times, and lack mammary glands and lungs. Still, the ease with which germ free Zebrafish can be produced in addition to their similar nervous system physiology make them a powerful model in the study of the microbiome [13].” The suggested citations were included.
Minor comments:
- Line 50 and throughout the text: Bifidobacterium. Comments: Please refer to
https://jcm.asm.org/content/nomenclature “Names of all taxa (kingdoms, phyla, classes, orders, families, genera, species, and subspecies) are printed in italics and should be italicized in the manuscript; strain designations and numbers are not”.
- Response: This has been addressed throughout the manuscript.
Sincerely,
Dr. Krzysztof Czaja

Round 2
Reviewer 1 Report
Minor revisions
- Page 1 line 29: microbiome does not need to be capitalized. (this is still capitalized from before)
- Figure 1: I think you need to have all the species in taxonomic format (like D. Melanogaster) or in common format (like rat, mice, zebrafish).
- Page 2 line 68: You need to define GF the first time you use the abbreviation.
- Page 3 line 89: Please delete the extra comma
- Page 3 line 109: Once you first define Drosophila melongaster, every subsequent reference should be D. melongaster. Please change throughout.
- Page 3 line 113: Zebrafish does not need to be capitalized. Please fix it throughout.
- Page 4 line 123: I don't necessarily agree with this transition as written. The above section doesn’t show me that the methodology is lacking, it shows me that there are a lot of options. I think you need to add something at the end of the zebrafish section that makes this transition flow better. Maybe you can mention that the species found in the zebrafish microbiome are completely different from the ones found in human and mouse guts.
- Page 4 line 132: You have a spacing issue here between the previous section and this section. Please fix
- Page 4 line 133: You have a ,. please delete the extra comma
- Page 4 line 152: SPE mice and conventional mice are sometimes different, check to see the exact one that was used in the study. Also if its SPE make sure to define this before using the abbreviation.
- Page 5 line 168: Lipopolysaccharide should not be capitalized.
- Page 5 line 175: You don't need to state that GF mice lack a microbiome because the reader knows that GF mice are germ-free.
- Page 5 line 215: Drosophila fly needs to be D. melongaster
- Page 10 line 443: you have the statement, “in people who have the J pouch surgery” and it looks like this was meant to be deleted
Major revisions
- There looks to be some font size issues in section 5, I would just go through and make sure everything is the same font size. Also, I think you need to go through and look at capitalizations. There were many unnecessary capitalizations (like Zebrafish, Microbiome, Lipopolysaccharide)
- Section two is almost there, but I think you need to work on more issues with the animal models since the point of this paragraph is to show how they are lacking and how they are not perfect. I think you hit the mouse model pretty good but the fruit fly and the zebrafish need more caveats associated with them.
I think this is really close to being good to go. The only section that gave me pause is the animal models section. Once that is cleaned up I will be happy to accept.
Author Response
Journal: Nutrients
Manuscript ID: nutrients-1040190
Title of paper: Managing the Microbiome: How the Gut Influences Development and Disease
Authors: Noah Weinstein, Brandon Garten, Jonathan Vainer, Dulce M. Minaya, Krzysztof Czaja *
Ms. Sarah Zhao,
We appreciate the time and efforts by the Editor and Referees in reviewing this manuscript. We have addressed all issues indicated in the review report (round 2) and believe that the revised version meets the journal’s publication requirements.
Minor revisions
- Page 1 line 28: microbiome does not need to be capitalized. (this is still capitalized from before)
- Response: This has been addressed.
- Figure 1: I think you need to have all the species in taxonomic format (like D. Melanogaster) or in common format (like rat, mice, zebrafish).
- Response: We decided to use common format, e.g., Fruit fly
- Page 2 line 71:You need to define GF the first time you use the abbreviation.
- Response: This has been addressed.
- Page 3 line 89: Please delete the extra comma
- Response: This has been addressed.
- Page 3 line 109: Once you first define Drosophila melongaster, every subsequent reference should be melongaster. Please change throughout.
- Response: This has been addressed.
- Page 3 line 113:Zebrafish does not need to be capitalized. Please fix it throughout.
- Response: This has been addressed.
- Page 4 line 123: I don't necessarily agree with this transition as written. The above section doesn’t show me that the methodology is lacking, it shows me that there are a lot of options. I think you need to add something at the end of the zebrafish section that makes this transition flow better. Maybe you can mention that the species found in the zebrafish microbiome are completely different from the ones found in human and mouse guts.
- Response: More emphasis has been placed on the discrepancies between human and zebrafish physiology, and the limitations these discrepancies impose on any study using the zebrafish model. Page 4 lines 123-140.
- Page 4 line 132:You have a spacing issue here between the previous section and this section. Please fix
- Response: All sections are spaced uniformly now.
- Page 4 line 133: You have a ,. please delete the extra comma
- Response: This has been addressed.
- Page 4 line 152: SPE mice and conventional mice are sometimes different, check to see the exact one that was used in the study. Also if its SPE make sure to define this before using the abbreviation.
- Response: SPF has been defined and confirmed as accurate.
- Page 5 line 177:Lipopolysaccharide should not be capitalized.
- Response: This has been addressed.
- Page 5 line 175:You don't need to state that GF mice lack a microbiome because the reader knows that GF mice are germ-free.
- Response: This has been addressed.
- Page 5 line 224: Drosophila fly needs to be melongaster
- Response: This has been addressed.
- Page 10 line 443:you have the statement, “in people who have the J pouch surgery” and it looks like this was meant to be deleted
- Response: This has been addressed.
Major revisions
- There looks to be some font size issues in section 5, I would just go through and make sure everything is the same font size. Also, I think you need to go through and look at capitalizations. There were many unnecessary capitalizations (like Zebrafish, Microbiome, Lipopolysaccharide)
- Response: We have gone through the manuscript to ensure style form is uniform throughout and unnecessary capitalizations have been corrected.
- Section two is almost there, but I think you need to work on more issues with the animal models since the point of this paragraph is to show how they are lacking and how they are not perfect. I think you hit the mouse model pretty good but the fruit fly and the zebrafish need more caveats associated with them.
- Response: We believe caveats of the fruit fly and zebrafish model have been more expressly presented and better integrated into the theme of this section as suggested. Page 4.
I think this is really close to being good to go. The only section that gave me pause is the animal models section. Once that is cleaned up I will be happy to accept.
Sincerely,
Dr. Krzysztof Czaja
